# Adherence to an Injury Prevention Warm-Up Program in Children’s Soccer—A Secondary Analysis of a Randomized Controlled Trial

**DOI:** 10.3390/ijerph182413134

**Published:** 2021-12-13

**Authors:** Matias Hilska, Mari Leppänen, Tommi Vasankari, Sari Aaltonen, Jani Raitanen, Anu M. Räisänen, Kathrin Steffen, Hannele Forsman, Niilo Konttinen, Urho M. Kujala, Kati Pasanen

**Affiliations:** 1Tampere Research Center of Sports Medicine, UKK Institute for Health Promotion Research, 33500 Tampere, Finland; mari.leppanen@ukkinstituutti.fi (M.L.); kati.pasanen@ucalgary.ca (K.P.); 2Tampere University Hospital, 33500 Tampere, Finland; 3UKK Institute for Health Promotion Research, 33500 Tampere, Finland; tommi.vasankari@ukkinstituutti.fi (T.V.); jani.raitanen@ukkinstituutti.fi (J.R.); 4Faculty of Medicine and Health Technology, Tampere University, 33520 Tampere, Finland; 5Institute for Molecular Medicine (FIMM), University of Helsinki, 00290 Helsinki, Finland; sari.s.aaltonen@helsinki.fi; 6Faculty of Social Sciences (Health Sciences), Tampere University, 33520 Tampere, Finland; 7Department of Physical Therapy Education—Oregon, College of Health Sciences—Northwest, Western University of Health Sciences, Lebanon, OR 97355, USA; araisanen@westernu.edu; 8Sport Injury Prevention Research Centre, Faculty of Kinesiology, University of Calgary, Calgary, AB T2N 1N4, Canada; 9Oslo Sports Trauma Research Center, Norwegian School of Sport Sciences, 0806 Oslo, Norway; kathrin.steffen@nih.no; 10Eerikkilä Sports Institute Training Center, Eerikkilä, 31370 Tammela, Finland; hannele.forsman@eerikkila.fi; 11Faculty of Sport and Health Sciences, University of Jyväskylä, 40014 Jyväskylä, Finland; niilo.konttinen@kihu.fi (N.K.); Urho.kujala@jyu.fi (U.M.K.); 12Research Institute for Olympic Sports, 40700 Jyväskylä, Finland; 13Alberta Children’s Hospital Research Institute, University of Calgary, Calgary, AB T2N 1N4, Canada; 14McCaig Institute for Bone and Joint Health, Cumming School of Medicine, University of Calgary, Calgary, AB T2N 1N4, Canada

**Keywords:** adherence, adolescent, children, football, implementation, injury prevention, neuromuscular training, soccer, youth

## Abstract

This study examined the impact of high adherence to a neuromuscular training (NMT) warm-up on the risk of lower extremity (LE) injuries in children’s soccer. Twenty U11–U14 youth clubs (*n* = 92 teams, 1409 players) were randomized into intervention (*n* = 44 teams) and control (*n* = 48 teams) groups. The intervention group was advised to perform an NMT warm-up 2 to 3 times a week for 20 weeks. Team adherence, injuries, and exposure were registered throughout the follow-up. Primary outcomes were the incidence of soccer-related acute LE injuries and the prevalence of overuse LE injuries. Intervention teams conducted mean 1.7 (SD 1.0) NMT warm-ups weekly through follow-up. The seasonal trend for adherence declined significantly by −1.9% (95% CI −0.8% to −3.1%) a week. There was no difference in the incidence of acute injuries nor the prevalence of overuse LE injuries in high team adherence group (*n* = 17 teams) compared to controls. However, the risk for acute noncontact LE injuries was 31% lower in the high team adherence group compared to controls (IRR 0.69, 95% CI 0.49 to 0.97). In an efficacy analysis (*n* = 7 teams), there was a significant reduction of 47% in the rate of noncontact LE injuries (IRR 0.53, 95% CI 0.29 to 0.97). In conclusion, teams conducted NMT warm-up sessions regularly, but with a declining trend. A greater protective effect was seen in teams with the highest adherence to the NMT warm-up.

## 1. Introduction

The injury preventive potential of neuromuscular training (NMT) in a randomized controlled trial (RCT) context has been well-established in youth soccer [1,2,3,4]. However, the effectiveness of injury preventive strategies depends most importantly on the implementation of these methods into everyday practice [5,6].

Adherence to the provided interventions is not only seen as a major modifying factor to injury risk in efficacy trials, but also as an individual outcome measure contributing to the effectiveness of the intervention [7,8]. Development of effective adherence strategies is called for and the first step is to identify the actual adherence rate of the methods in use [9].

In soccer, two NMT warm-up sessions a week has been proposed as a sufficient rate for successful injury prevention effect [10]. There is, however, a limited number of prospective studies examining adherence to NMT in children. Two studies in U13 children’s soccer have reported average weekly NMT warm-up sessions of 1.0 and 1.9 per week, but without closer examination of the development of conducted weekly NMT warm-up sessions during follow-up (trend of adherence) [2,4]. Four studies in female U13–U18 youth soccer have reported adherence to an NMT program in detail [11,12,13,14]. Three of these studies reported the weekly number of sessions between 1.0–1.4 and a significant declining trend during a follow-up [11,12,14], whereas one study reported good adherence through follow-up with a mean >2 sessions per week [13].

At the same time, high adherence to prevention programs have shown to provide the greatest protective effect from injuries [2,4,12,13,14,15]. The definition of high adherence is based on a tertile split of the participants into high, medium, and low adherence groups in most of the studies [2,4,12,13,14]. In the U13 children’s studies, players in the high adherence group were at 64% and 72% lower risk for overall injuries, while no effect was seen in low adherence groups compared to controls [2,4]. Soligard et al. reported players with high adherence to an NMT warm-up being at a 35% lower risk for injuries than players with medium adherence to the intervention [12]. Steffen et al. found an even greater difference of 72% lower injury risk in a similar comparison [13]. A study examining anterior cruciate ligament knee injuries specifically reported an 88% risk decrease in high adherent individuals to an NMT program compared to controls without any intervention [14].

There is a limitation in interpreting these study results as all but one [2] focus on individual player adherence, not team adherence. Player adherence represents how many NMT warm-up sessions individual players complete while team adherence represents how many NMT warm-up sessions teams organize for their players. Time-loss injuries lead to temporary or complete cessation in sport participation, decreasing individual participant’s adherence to intervention training. Thus, players who have not sustained injuries are more likely to have a higher adherence to the examined intervention and are overrepresented in the high adherence group. This may result in overestimation of intervention efficacy. Therefore, we propose team adherence to be a more valid indicator of the effect of adherence to the efficacy of an intervention.

The primary aim of this study was to evaluate the adherence to an NMT warm-up program in children’s U11–U14 soccer teams and to examine whether a high adherence to an NMT warm-up program can prevent acute and overuse lower extremity (LE) injuries. A secondary aim was to descriptively examine the intervention team coaches’ attitudes towards and maintenance of the NMT warm-up in weekly practice.

## 2. Materials and Methods

This is a secondary analysis of data from a RCT that investigated the impact of an NMT warm-up on acute and overuse LE injuries in children’s soccer (ISRCTN14046021). The study design followed the Consolidating Standards for Reporting Clinical Trials (CONSORT) [16] and is reported in detail elsewhere [17,18]. The Ethics Committee of Pirkanmaa Hospital District (ETL-code R13110) approved the study.

### 2.1. Participants and Randomization

The Sami Hyypiä Academy (SHA) is a national training and research center for Finnish soccer. The SHA involves top-level youth soccer clubs nationwide to participate in a player monitoring program. All 20 SHA clubs (92 teams, *n* = 1643 players) were invited to participate in this study (Figure 1). All official players of the participating U11–U14 teams were eligible to enter the study, regardless of their injury status at baseline. Players joining the teams after the study commencement were not included. Written consent was obtained from each participant and their guardian at enrollment. Baseline data were collected in monitoring events in fall 2014. The clubs were randomly allocated to the intervention and control groups with the home city (*n* = 13) of a club as the unit of randomization.

### 2.2. Intervention

The intervention was carried out during Spring 2015 (in total, 20 weeks) including both pre-season (January–March) and competitive season (April–June). Coaches from the intervention group were provided with one pre-season coach workshop including both theoretical and practical sessions on injury prevention in children’s soccer and detailed description and demonstration of the NMT warm-up program. In addition, each coach received a tablet computer including written instructions and videos of each exercise to support the execution of the program. During follow-up, researchers visited each intervention team 2–3 times to oversee intervention.

The NMT warm-up was designed to improve players’ movement control and prevent LE injuries, and was based on the researchers’ previous injury prevention NMT warm-up [15]. The program could be modified from practice to practice as it consisted of seven different exercises with progression and variations of diverse difficulty (Appendix A). Teams in the intervention group were instructed to replace standard warm-up with the NMT warm-up before soccer practices 2–3 times a week and 20 min each.

Control teams were instructed to continue their training routines as usual. Control teams knew to be the control arm of a training intervention study and expected to receive the same workshop as the intervention teams after the study. Researchers conducted 1–2 check-ups to the control teams’ practices during the follow-up.

### 2.3. Team and Player Adherence

Adherence to the NMT program was recorded by team coaches. They were asked to keep track of each conducted NMT warm-up session and attendance of each individual player in a structured online form.

Adherence was defined as a four-level categorical variable (controls; low-, medium-, and high-adherence groups) and was examined at both team and individual levels. Team adherence groups were stratified based on the researchers’ clinical judgement of how the warm-up was conducted during follow-up on a weekly basis (Appendix A). This evaluation resulted in cut-offs of low adherence: under 25 NMT warm-up sessions during follow-up; medium: 25 to 35; and high > 35. An additional efficacy analysis was made for the teams (*n* = 7) that adhered ideally per protocol to the program (≥40 NMT warm-up sessions translating to two sessions a week and no more than two single-week breaks during the 20-week intervention period). At an individual level, the participants were stratified into tertiles of adherence (low, medium, high) according to completed NMT warm-up sessions during follow-up [4,12,13,14].

### 2.4. Injury and Exposure Recording

Injuries were defined based on the 2006 consensus statement [19]. For injury, the definition “any physical complaint sustained by a player that resulted from football training or playing, irrespective of the need for medical attention or time-loss from football activities” was used. For acute injury, “an injury resulting from a specific, identifiable event”; and for overuse injury, “an injury caused by repeated microtrauma without a single identifiable event responsible for the injury” were used. Noncontact injuries and the anatomical location of the injury was categorized following the same consensus statement. Severity of overuse injuries were defined substantial following validation guidelines [20].

Injuries were tracked weekly using a short message service (SMS) approach. Every Sunday, all guardians received the following SMS: Has your child had any musculoskeletal complaint or injuries during the previous seven days (yes/no)? If a guardian did not reply to the SMS, they received a reminder SMS during the following week. Four blinded study physiotherapists contacted guardians who had reported an injury and completed a structured 10-min telephone interview. For overuse injuries, the OSTRC-O questionnaire was used [20].

Weekly exposure to soccer practice and games were self-reported by the players using the SHA’s internet-based player monitoring system. The players were asked to submit their exposure form once a month. These reports were incomplete and instead of individual exposure, team-based exposure data were used in the analysis. Team mean weekly practice exposure was calculated from data returned by individual team players and was applied to all team members. Game exposure data was retrieved from the SHA for wintertime pre-season (January–March) and from the Finnish Football Association for the competitive season (April–June). Game exposure was derived from standard game durations for each age group and number of players on the field and applied for entire teams [19].

### 2.5. Coach Attitudes and Maintenance of the Program

After the intervention, a questionnaire survey was conducted for both intervention and control team coaches. Control team coaches were asked to describe their warm-up training routines and possible unprompted injury preventive measures used during follow-up. Intervention team coaches were asked to complete a questionnaire to evaluate their experiences and attitudes to the injury prevention NMT warm-up. Furthermore, a questionnaire for the intervention team coaches was conducted six months after the intervention period to evaluate adoption and maintenance of the NMT warm-up after the study.

### 2.6. Statistics

Statistical analyses were performed using Stata Statistical Software V15.0 (StataCorp, College Station, TX, USA) and IBM SPSS V27.0 (IBM Corp., Armonk, NY, USA). Confidence intervals (CI) for injury incidences and prevalences were calculated with the OpenEpi V3 statistical calculator using Mid-*p* Exact test.

Participant characteristics were compared between groups of adherence and the controls with one-way analysis of variance and chi-square test where appropriate. The results are expressed as means with standard deviations (SD) or the number (*n*) within group.

Seasonal trend for adherence of the entire intervention group was analyzed using a generalized linear mixed model with a negative binomial distribution. The number of weekly NMT warm-up sessions was set as the outcome variable and the follow-up week as the explanatory variable. The team was used as random effect. The results are expressed as a relative weekly change (%) in the number of conducted NMT warm-up sessions with 95% CIs. Mean weekly sessions by month (defined as 4-week periods) in the adherence groups are shown for illustration.

For injuries, acute injury incidence per 1000 h of soccer practice and games and weekly overuse injury prevalence between groups were compared. Noncontact acute injuries and substantial overuse injuries were analyzed supplementally. Differences between the groups were analyzed with negative binomial regression for acute injury incidence and with a generalized linear mixed model for overuse injury prevalence. Cluster-robust standard errors were calculated to adjust for intragroup correlation between players in the same clubs. Adjustments were made for age and sex (for acute injuries) and age, sex, and exposure hours (for overuse injuries) in a Supplementary Analysis. The results are expressed as incidence rate ratios (IRR) for acute injuries and odds ratios (OR) for overuse injuries together with 95% CIs.

## 3. Results

### 3.1. Participants

All recruited clubs agreed to participate, but 219 participants declined, resulting in 1424 players. Fifteen players stopped playing in the participating teams between study recruitment and start of follow-up. Thus, the study population consisted of 1409 players randomized into intervention (*n* = 676 players) and control (*n* = 733) groups. Drop-out rate during follow-up was 4% (56 players) (Figure 1). The average response rate to the weekly injury data collection was 95%, and 72% of the players answered every week.

Allocation of intervention teams into high-, medium-, and low-adherence groups resulted in dissimilar groups according to sex and age distribution, previous injuries sustained, and weekly soccer exposure hours during follow-up. Weekly soccer exposure was the highest in the group adhering lowest to the intervention (Table 1).

### 3.2. Adherence

All intervention teams reported adherence data. Individual adherence was obtained from 618 out of 676 players (91%).

A median 33 (range 4–57) NMT warm-up sessions per team were conducted during the 20-week follow-up corresponding to a median 2 (0–5) sessions per week. Mean attendance of players participating in NMT warm-up sessions was 71% through follow-up. Two teams from a single club reported only 4 and 13 training sessions and withdrew from the intervention program protocol after starting their own physiotherapist-supervised injury prevention program in the second month of follow-up. All other teams (42/44) completed more than 20 training sessions and followed the intervention regularly with ≥1 training sessions on 15/20 follow-up weeks (Appendix A).

The number of conducted NMT warm-up sessions in the intervention group declined significantly each week by −1.9% (95% CI −0.8% to −3.1%) from mean 2.0 weekly NMT warm-up sessions in the first month of follow-up to 1.5 in the last month of follow-up. The trend for adherence in different adherence groups is illustrated in Figure 2 and Figure 3.

### 3.3. Acute Injuries

The incidence of acute LE injuries was 4.77 per 1000 h of exposure in the high team adherence group and 5.48 in the control group and there was no statistical difference between the groups. However, the low team adherence group’s LE injury incidence (3.50) was significantly lower compared to the control group: IRR 0.66 (95% CI 0.45 to 0.96) (Table 2).

In individual level adherence, players of high adherence were at significantly lower risk for acute LE injuries than the controls: IRR 0.77 (95% CI 0.61 to 0.96).

In comparison to the controls, the incidence of acute noncontact LE injuries was significantly lower in the high (IRR 0.69, 95% CI 0.49 to 0.97) and low (IRR 0.53, 95% CI 0.36 to 0.77) team adherence groups, but not in the medium (IRR 0.69, 95% CI 0.46 to 1.02) team adherence group.

### 3.4. Overuse Injuries

Examining team adherence, the mean weekly prevalence of overuse LE injuries was 12.5% (95% CI 11.6 to 13.5) in high adherence group, 10.2% (9.4 to 11.0) in medium adherence group, 15.5% (13.2 to 18.0) in low adherence group, and 11.3% (10.7 to 11.9) in the control group. There were no differences in the prevalence of overuse LE injuries between the adherence groups compared to the control group. No differences in the prevalence of overuse LE injuries was seen at an individual level either (Table 3).

### 3.5. Efficacy Analysis

Seven teams adopted the intervention program ideally per protocol during the follow-up. The injury incidence was 4.72 per 1000 h of exposure for all LE injuries and 1.45 for noncontact LE injuries. Noncontact LE injury incidence was significantly lower when compared to the controls: IRR 0.53 (95% CI 0.29 to 0.97). The mean weekly prevalence of overuse LE injuries was 13.2% in these ideally adhering teams and this was similar to the control group’s weekly overuse LE injury prevalence (Table 4).

### 3.6. Coach Attitudes and Maintenance of the Program

At the end of the intervention, a total 35 out of 44 intervention team coaches completed a questionnaire considering the attitudes and beliefs towards the NMT warm-up. All intervention team coaches answering considered the NMT warm-up beneficial. In addition, 22 intervention team coaches completed the survey to evaluate adoption and maintenance of the NMT warm-up program six months after the intervention: 18 out of 22 teams (82%) were still conducting the exercises of the study NMT warm-up in some composition. Open feedback was received from 25 coaches, and the most relevant feedback of the NMT warm-up is listed as follows:The warm-up was too long in duration. (*n* = 3)Overuse injury complaints were experienced to increase. (*n* = 2)The fidelity of performing the exercises varied between players. (*n* = 2)There were problems with limited spaces and limited time to do the warm-up properly. (*n* = 2)The involvement of ball skills in the warm-up was too little. (*n* = 1)Some exercises were too difficult for those under 12-years old. (*n* = 1)The warm-up was too demanding and tiring for the players in the beginning. (*n* = 1)

## 4. Discussion

We analyzed adherence to an NMT warm-up and its impact on LE injury risk in a clinical trial setting in children’s soccer. The teams reached target adherence of at least 2 sessions per week insufficiently, but still conducted regular training with mean 1.7 sessions per week. High adherence was found to prevent acute noncontact LE injuries.

Overall, we saw a minor decreasing trend in conducted NMT sessions through the follow-up. However, excluding the first month of higher adherence to the intervention, the number of weekly sessions was consistent for rest of the follow up. The differences in adherence between most of the teams were small and it was difficult to differentiate between the high-, medium-, and low-adherence groups. These findings implicate good potential for the real-life implementation of NMT warm-up programs in children’s soccer. Additionally, the observed adherence in our study is in line with the earlier studies in U13 and U13–U18 soccer, where weekly adherence has averaged between 1 to 2 NMT warm-ups per week [2,4,12,13,14].

The NMT warm-up did not prove to be effective in preventing all acute LE injuries nor overuse LE injuries in the teams of high adherence. However, a 31% decreased risk in acute noncontact LE injuries was seen in the high adherence teams but not in the medium adherence teams. Furthermore, at the individual level, players with high adherence to the intervention were at 23% lower risk for all acute LE injuries than the controls, and again, this was not seen in players of medium adherence to the intervention. The results indicate that regular adherence to NMT warm-up is required for a protective effect against injuries and this contributes to the existing evidence in U13 and U13–U18 soccer, where regular adherence has also proven to increase the injury preventive effect of NMT warm-up [2,4,12,13,14].

The acute LE injury risk was also decreased in the low team adherence intervention group. Two of the four teams in low adherence group were from the same club, and the reason for nonadherence in this club was the use of their own similar injury prevention warm-up designed and supervised by their team physiotherapist. It seems that there were no true low adherence teams in our study.

Successful implementation of injury prevention programs from controlled trials to weekly practice is a complex task [6,9]. Our approach was intended to be minimally invasive: we made an effort to mimic real-life settings as the intervention consisted of coach workshop at baseline and a tablet computer to support the execution of the intervention. A similar approach from an earlier study was considered low-level intervention in contrast to comprehensive intervention [13]. We chose these methods with an aim to facilitate the translation of the NMT warm-up from study setting to a wider implementation outside of the study. All intervention team coaches answering to the post-study survey considered the NMT warm-up beneficial, and six months after the study, 82% of the intervention teams who replied reported using components of the NMT warm-up regularly in their training. An earlier implementation study reported a similar rate (80%) of their RCT’s intervention group coaches using the studied NMT warm-up three years after the study [21]. These results of maintaining the behavior after the controlled study are encouraging.

Future injury prevention research should continue to facilitate the adoption and maintenance of proven intervention measures into real-life use. Sports clubs, associations, and community groups are key in the implementation process and we would advise organizations to regularly educate children’s soccer coaches in injury prevention [22,23,24]. Similar physiotherapist-led coach workshops, as used in our study, would be a cost-efficient way to deliver information and practical skills straight to the soccer coaches in the field. When possible, designating warm-ups to an assistant coach in the teams would likely improve the fidelity of the exercises.

Team coaches are willing to modify the existing NMT warm-ups for their own preferences [21,25,26]. In our experience from children’s soccer, these modifications most often concern including ball skills in the warm-up, introducing exercises sporadically into practice, or limiting the overall duration of the comprehensive program. Co-operating with the team coaches and other program deliverers is key, and involving ball skills would be wise in aim to improve the adoption of NMT exercises regularly into practice. However, proper modification requires informed decision making and regular injury prevention education; NMT warm-up needs to be systematic and regular in weekly use, the exercises done with fidelity and the modifications should not leave out any components of the exercise categories in the long-term. Emphasis should be made to those exercises which are taken less readily into maintained use from controlled trial settings.

### 4.1. Strengths and Limitations

The intervention was operated by team coaches who were responsible for their team’s everyday training irrespective of the study setting. Study personnel only visited the teams infrequently to observe training. The method builds for external validity of the results, and for the transferability of the intervention to real-life community-level implementation. In fact, we conducted a post-study survey and components of the NMT warm-up were maintained well. However, these results would have been reasonable to confirm with visits to the clubs and observing the teams’ warm-up routines in practice, and not only by interviewing the coaches as was conducted.

The study sample was large and representative of competitive 9–14-year-old soccer players all over Finland. Females were in a minority, but this primarily reflects the sex distribution of competitive youth players in Finland, as all female clubs from the SHA were recruited. Compliance to the injury data collection was excellent and only a few subjects dropped out of the study.

Soccer exposure varied in the different groups of adherence and the controls, and must be considered a confounding factor. The intervention teams had more weekly exposure than the controls, contributing to a higher training load and a possible increase in injury risk [27,28]. The intervention teams may have conducted the NMT warm-up in addition to their regular training despite being instructed to replace regular warm-up, not to lengthen their training sessions. Furthermore, the missing of individual exposure data and use of team-based exposure instead is a limitation of the study.

The documentation of NMT warm-up sessions was good and made in detail by the team coaches. However, as team practices were not observed regularly during follow-up, no assessment for exercise fidelity—as was examined in two previous studies—was undertaken [29,30]. Similar to the objective results of these two studies, some intervention team coaches in our study pointed out that their players conducted the exercises with varying technique—an important factor to emphasize in the delivery of NMT interventions.

### 4.2. Clinical Implementation

Children’s soccer teams adhered moderately to the designed NMT warm-up and components of the NMT warm-up were adopted for use after the controlled trial setting. It seems that conducting NMT warm-up twice a week regularly is needed for effective acute noncontact LE injury prevention. Communities and sporting organizations should educate youth soccer coaches and health care professionals regularly in injury prevention with NMT and encourage them to adopt it as part of their weekly practice routines.

## 5. Conclusions

High adherence from teams to an NMT warm-up prevents acute noncontact LE injuries, but it has no impact on the prevalence of overuse LE injuries. A threshold of sufficient adherence is needed for an injury preventive effect to be seen and a minimum of two NMT warm-up sessions a week is recommended. The studied NMT warm-up has good potential to be adopted in children’s weekly soccer practice.

## Figures and Tables

**Figure 1 ijerph-18-13134-f001:**
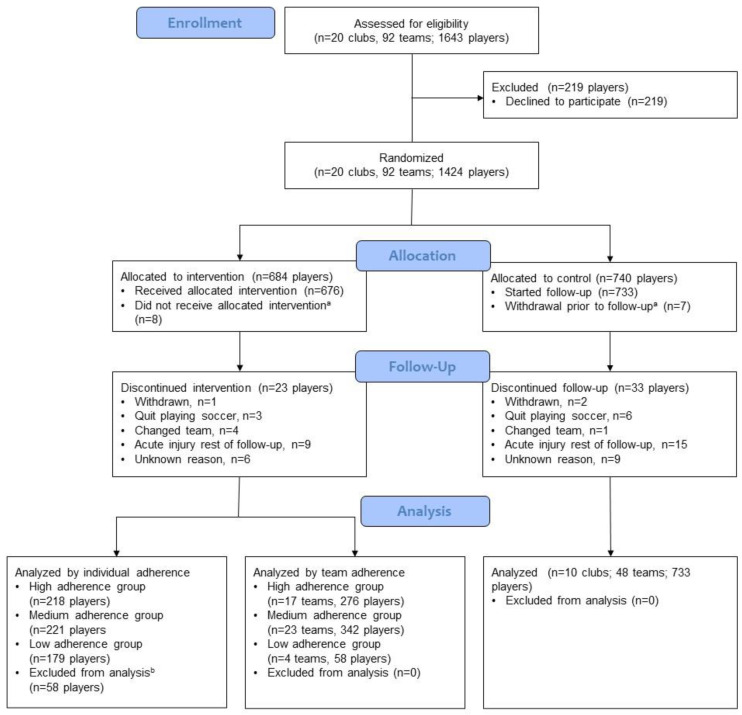
CONSORT Flow Chart. ^a^ Stopped playing in the participating teams prior to follow-up. ^b^ Missing individual adherence data.

**Figure 2 ijerph-18-13134-f002:**
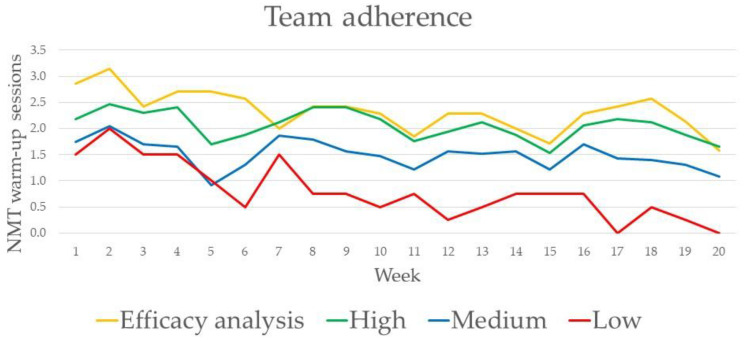
Trend in the adherence to a neuromuscular training (NMT) warm-up through 20-week follow-up. Each line represents mean weekly NMT warm-up sessions within the group; Number of teams: Efficacy analysis (*n* = 7 teams), High (*n* = 17), Medium (*n* = 23), Low (*n* = 4).

**Figure 3 ijerph-18-13134-f003:**
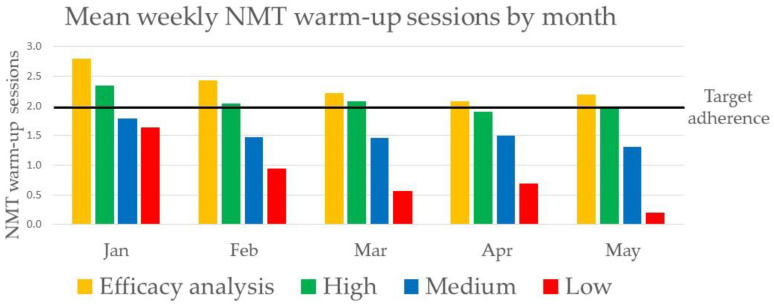
Mean weekly neuromuscular training (NMT) warm-up sessions by month in the different groups of adherence to the intervention. Number of teams: efficacy analysis (*n* = 7 teams), high (*n* = 17), medium (*n* = 23), low (*n* = 4).

**Table 1 ijerph-18-13134-t001:** Player characteristics by team adherence groups.

	High	Medium	Low	Control	*p*-Value ^1^
Teams, *n*	17	23	4	48	-
Players, *n*	276	342	58	733	-
Females, %	15	21	5	22	0.002
Age, mean (SD)	12.2 (1.3)	12.0 (1.1)	13.2 (0.8)	12.3 (1.1)	<0.001
Previous injuries (yes/no) ^2^, yes *n* (%)	124 (45)	141 (41)	34 (59)	291 (40)	0.002
Weekly exposure hours, mean (SD)	5.1 (1.5)	5.4 (1.2)	7.2 (1.2)	4.5 (0.9)	<0.001
NMT warm-up sessions per week, mean (SD)	2.1 (0.3)	1.5 (0.2)	0.8 (0.5)	-	<0.001

^1^ One-way analysis of variance for continuous data and chi-square test for categorical data; ^2^ During the previous 12 months. Data from 1147 players.

**Table 2 ijerph-18-13134-t002:** Negative binomial regression analysis of acute lower extremity (LE) injury incidence between intervention adherence groups compared to control group.

	Team Adherence				Player Adherence		
	Injury Incidence per 1000 h (95% CI)	Crude ^1^ IRR (95% CI)	Adjusted ^2^ IRR (95% CI)		Injury Incidence per 1000 h	Crude ^1^ IRR (95% CI)	Adjusted ^2^ IRR (95% CI)
**LE Injuries**							
High	4.77 (4.01 to 5.64)	0.87 (0.71 to 1.08)	0.88 (0.71 to 1.10)	High	4.15 (3.37 to 5.06)	**0.76 (0.60 to 0.96)**	**0.77 (0.61 to 0.96)**
Medium	4.17 (3.54 to 4.88)	0.77 (0.54 to 1.09)	0.78 (0.57 to 1.08)	Medium	5.14 (4.29 to 6.12)	0.96 (0.72 to 1.27)	0.97 (0.73 to 1.28)
Low	3.50 (2.35 to 5.02)	**0.66 (0.48 to 0.90)**	**0.66 (0.45 to 0.96)**	Low	4.23 (3.38 to 5.23)	0.78 (0.55 to 1.11)	0.79 (0.57 to 1.11)
Control	5.48 (4.93 to 6.08)	1 (reference)	1 (reference)	Control	5.48 (4.93 to 6.08)	1 (reference)	1 (reference)
**Noncontact LE Injuries**							
High	1.87 (1.41 to 2.43)	**0.67 (0.48 to 0.94)**	**0.69 (0.49 to 0.97)**	High	1.87 (1.37 to 2.51)	**0.67 (0.51 to 0.89)**	**0.68 (0.51 to 0.91)**
Medium	1.83 (1.43 to 2.32)	0.67 (0.42 to 1.07)	0.69 (0.46 to 1.02)	Medium	2.09 (1.57 to 2.73)	0.77 (0.47 to 1.24)	0.79 (0.49 to 1.25)
Low	1.43 (0.75 to 2.48)	**0.52 (0.34 to 0.79)**	**0.53 (0.36 to 0.77)**	Low	1.67 (1.16 to 2.33)	0.61 (0.34 to 1.08)	0.63 (0.37 to 1.05)
Control	2.76 (2.38 to 3.20)	1 (reference)	1 (reference)	Control	2.76 (2.38 to 3.20)	1 (reference)	1 (reference)

IRR = Incidence rate ratio; CI = Confidence interval; Significant results are in **bold**. ^1^ Adjusted for soccer exposure hours; ^2^ Adjusted for soccer exposure hours, age, and sex.

**Table 3 ijerph-18-13134-t003:** Generalized linear mixed model analysis of overuse lower extremity (LE) injury prevalence between intervention adherence groups compared to control group.

		Team Adherence				Player Adherence		
		Mean Weekly Injury Prevalence % (95% CI)	Crude ^1^ OR (95% CI)	Adjusted ^2^ OR		Mean Weekly Injury Prevalence % (95% CI)	Crude ^1^ OR (95% CI)	Adjusted ^2^ OR
**LE Injuries**								
	High	12.5 (11.6 to 13.5)	1.01 (0.98 to 1.04)	1.01 (0.98 to 1.04)	High	8.9 (8.0 to 9.8)	1.01 (0.98 to 1.05)	1.01 (0.98 to 1.05)
	Medium	10.2 (9.4 to 11.0)	1.01 (0.98 to 1.04)	1.01 (0.98 to 1.04)	Medium	9.3 (8.4 to 10.2)	1.00 (0.97 to 1.03)	1.00 (0.97 to 1.03)
	Low	15.5 (13.2 to 18.0)	1.01 (0.96 to 1.07)	1.01 (0.96 to 1.07)	Low	15.5 (14.3 to 16.9)	1.02 (0.99 to 1.05)	1.02 (0.99 to 1.05)
	Control	11.3 (10.7 to 11.9)	1 (reference)	1 (reference)	Control	11.3 (10.7 to 11.9)	1 (reference)	1 (reference)
**Substantial LE Injuries**								
	High	6.1 (5.5 to 6.8)	1.00 (0.96 to 1.04)	1.00 (0.96 to 1.04)	High	3.4 (2.9 to 4.0)	1.00 (0.95 to 1.05)	1.00 (0.95 to 1.05)
	Medium	5.5 (4.9 to 6.1)	0.99 (0.96 to 1.03)	0.99 (0.96 to 1.03)	Medium	4.6 (4.0 to 5.3)	0.99 (0.94 to 1.03)	0.99 (0.94 to 1.03)
	Low	9.2 (7.5 to 11.2)	1.00 (0.94 to 1.07)	1.01 (0.94 to 1.08)	Low	8.6 (7.7 to 9.6)	1.01 (0.97 to 1.05)	1.01 (0.97 to 1.05)
	Control	5.0 (4.7 to 5.4)	1 (reference)	1 (reference)	Control	5.0 (4.7 to 5.4)	1 (reference)	1 (reference)

OR = odds ratio; CI = confidence interval; ^1^ Unadjusted for any variables. ^2^ Adjusted for soccer exposure hours, age, and sex.

**Table 4 ijerph-18-13134-t004:** Efficacy analysis of acute lower extremity (LE) injury incidence and overuse LE injury prevalence in ideal adherence intervention group compared to control group.

**Acute Injuries**	**Injury Incidence per 1000 h (95% CI)**	**Crude ^1^ IRR** **(95% CI)**	**Adjusted ^2^ IRR** **(95% CI)**
LE injuries	4.72 (3.56 to 6.14)	0.87 (0.57 to 1.34)	0.87 (0.59 to 1.29)
Noncontact LE injuries	1.45 (0.86 to 2.31)	0.52 (0.28 to 1.001)	**0.53 (0.29 to 0.97)**
**Overuse Injuries**	**Mean Weekly Injury** **Prevalence** **% (95% CI)**	**Crude ^3^ OR** **(95% CI)**	**Adjusted ^4^ OR** **(95% CI)**
LE injuries	13.2 (11.8 to 14.7)	0.99 (0.95 to 1.03)	0.99 (0.95 to 1.03)
Substantial LE injuries	6.3 (5.4 to 7.4)	0.98 (0.93 to 1.03)	0.98 (0.93 to 1.03)

IRR = incidence rate ratio; OR = odds ratio; CI = confidence interval; Comparisons (IRR’s and OR’s) are made in reference to the control group. Significant results are in **bold**. ^1^ Adjusted for soccer exposure hours. ^2^ Adjusted for soccer exposure hours, age and sex. ^3^ Unadjusted for any variables. ^4^ Adjusted for soccer exposure hours, age, and sex.

## Data Availability

Unpublished data are available upon request according to trial protocol.

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
