# Peer review of "Adherence to an Injury Prevention Warm-Up Program in Children’s Soccer—A Secondary Analysis of a Randomized Controlled Trial"

_ijerph, 2021, doi:10.3390/ijerph182413134_

Round 1
Reviewer 1 Report
This is an interesting study that provides important insight for injury prevention implementation in children's sport.
A few comments for consideration:
Introduction:
In this section can the authors identify a definition(s) used in previous studies that demonstrate what high/medium/low adherences refers to and make comment on whether this definition is consistent/similar to those definitions used in the present study? The latter may need to appear in the discussion. The authors make references to their overall observed adherence as being in line with others studies, but a comparison of categories of adherence would also be valuable for clarity for the reader.
Line 57 - require full stop after 'adherence [2,4]'
Line 60 - capital 'W' for "whereas"
Line 62 - suggest deleting 'the' before the word 'prevention'
Line 64 - use the word 'risk' rather than 'risks'
Line 67 - full stop after '[12]'
Line 68 - fix reference format (13) and move full stop
Line 69 - suggest changing sentence to read: "examining anterior cruciate ligament knee injuries, specifically, reported an 88%..."
Line 71 - remove word 'However' and begin new paragraph on line 72 starting with 'There'. Also adjust sentence to read: "interpreting these study results, as all but one study..."
Line 79 - add 'primary' to read: "The primary aim..."
Line 81 - change sentence to read: "A secondary aim was to examine the intervention team coaches'..."
Material and Methods:
Line 96 - change word 'start' to "commencement"
Line 127 - change word 'in' to "at"
Lines 127-131 - refer to the earlier comment about identifying defined categories in previous studies and compare t the present.
Line 129 - change 'in weekly' to "on a weekly". Move full stop to after bracket text.
Line 134 - change word 'In' to "At an"
Line 136 - move full stop to after refs
Line 143 - add "were used" after definition for overuse injury.
Line 151 - can you provide a duration for how long the structured telephone interview was here?
Line 156 - add comma after 'individual exposure' and change wording 'data was' to "data were"
Line 161 - remove comma after 'field'
Results:
Line 239 - add space between words 'different and groups' in figure title
Line 242 - reference in text is to injury 'rate' not incidence, therefore suggest modification to text accordingly. This is also the case in Line 244.
Line 283 - change 'was' to "as"
Discussion:
Lines 359-360 - suggest clarifying wording
Line 366 - replace 'we did' with "was conducted"
Line 376 - change wording to read: "regular training despite being instructed to replace regular warm-up, ..."
Line 380 - change wording to read: "However, as team practices were not observed regularly during follow-up, no assessment for exercise fidelity, as was examined in two previous studies, was undertaken".
One final comment relates to the authors providing a very brief description of when/how follow-up occurred. Obviously the specific details of the methods are published elsewhere, however it would be useful to have some brief information about the follow up to provide clear context for data collection.
Reviewer 2 Report
The effect of NMT warm-up training on the risk of LE injury in children's football players is the main question addressed by the research. The topic original is relevant in the field. Compared with other published material, It provides data for both sexes in a certain age group in a wide subject group.
The methodology should be designed from the beginning and carried out between the consultant groups. For example, all groups should have football-specific training models and exposure times that coincide. In general, this study has issues with the consistency of the study groups and study design.
Reviewer 3 Report
I would like to commend the authors on such a large scale project in a topic which is highly contemporary. However, there a number of areas that I feel would require amendement before publication.
- The inclusion of the coaches perceptions of the programme and adherence issues is not within the context of the title and while the information is valuable to understand, this paper is not the best place for doing so. The paper would be more concise and focused if these elements were removed and reported in an additional study.
- I have some concerns regarding the classification of injury and the inclusion of all in injuries based on a self-reporting mechanism. The data would be more robust if these were reported as time-loss injuries.
- The writing of the discussion is somewhat conversational and includes many statements which are vague and do not communicate the authors points in a clear and concise manner, lines 304-308 for example.
Reviewer 4 Report
The present study aimed to evaluate the adherence to a neuromuscular training warm-up program in U11-U14 soccer teams and examine whether a high adherence can prevent acute and overuse lower extremity injuries. A secondary purpose was to analyze the coaches’ attitudes towards and maintenance of the warm-up in weekly practice. Teams conducted warm-up sessions regularly but with a declining trend. The Authors found no difference in the incidence of acute nor the prevalence of overuse injuries in the high team adherence group compared to controls. Furthermore, the risk for acute noncontact injuries was lower in the high team adherence group compared to controls.
Overall, the study is well designed and scientifically sound, and I compliment the Authors for their excellent work. The provided background is solid, although some more references are needed. The methods section is complete, but in some parts, a better procedures explanation would help the readers fully understand them, as well as the results where the interpretation (given the different analyses and outcomes) is sometimes challenging. I think excessive use of acronyms should be avoided to facilitate reading, please leave only the strictly necessary ones.
Minor specific comments are provided below.
Title: in my opinion, “a secondary analysis” could be omitted from the title since it does not alter the title meaning, and it is also specified in the methods section
Line 57: please define what “trend in adherence” represents.
Line 72: at this introductory point, the difference between individual and team adherence is not clear yet. Please explain to provide to the readers a complete background.
Lines 72-77: please provide some references to support these three statements.
Lines 81-83: the secondary aim seems vague. Please clarify briefly, explaining how “attitudes” and “maintenance” have been quantified.
Line 95: why were age groups restricted to U11-U14 in this study? Please explain.
Line 107: it could be helpful to provide a concise explanation of the contents of the workshop provided to coaches.
Line 155 and 160: both team-based exposure data and standard game durations could not precisely represent the individual practice and games exposure. Please provide a rationale for these indirect measures.
Line 339: please provide practical examples on how “facilitate the adoption and maintenance” of the described intervention measures.
